# Non-Mydriatic Ultra-Wide Field Imaging Versus Dilated Fundus Exam and Intraoperative Findings for Assessment of Rhegmatogenous Retinal Detachment

**DOI:** 10.3390/brainsci10080521

**Published:** 2020-08-05

**Authors:** Beatriz Abadia, Maria Carmen Desco, Jorge Mataix, Elena Palacios, Amparo Navea, Pilar Calvo, Antonio Ferreras

**Affiliations:** 1Clínica Baviera, 50001 Zaragoza, Spain; b.abadia@hotmail.com (B.A.); xenatrance@yahoo.es (P.C.); 2FISABIO-Oftalmología (FOM), 46035 Valencia, Spain; carmen.desco@uv.es (M.C.D.); jorge.mataix@gmail.com (J.M.); epalaciospozo@gmail.com (E.P.);; 3Instituto de la Retina, 46005 Valencia, Spain; amparo.navea@gmail.com; 4IIS-Aragón, Miguel Servet University Hospital, 50009 Zaragoza, Spain; 5Department of Surgery, University of Zaragoza, 50009 Zaragoza, Spain

**Keywords:** imaging, ophthalmoscopy, retina, retinal detachment, ultra-wide field imaging

## Abstract

Background: to compare the extent of the detached retina and retinal tears location in rhegmatogenous retinal detachment (RRD) among non-mydriatic ultra-wide field (UWF) imaging, dilated fundus exam (DFE), and intraoperative evaluation. Methods: this retrospective chart review comprised 123 patients undergoing surgery for RRD. A masked retina specialist analyzed the UWF fundus images for RRD area, status of the macula, and presence and location of retinal breaks. The same variables were collected from a database including DFE and intraoperative recordings. Evaluation methods were compared. Results: mean age was 59.8 ± 14.9 years. Best-corrected visual acuity improved from 0.25 ± 0.3 (Snellen) to 0.67 ± 0.3 at 12 months (p = 0.009). The RRD description and assessment of macula status (34.5% macula-on) did not differ between UWF, DFE, and intraoperative examination. The inferior quadrant was involved most frequently (41.5%), followed by the superior (38.9%), temporal (27.8%) and nasal quadrant (14.8%). Intraoperative exam detected 96.7% of retinal tears compared with DFE (73.2%, p = 0.008) and UWF imaging (65%, p=0.003). UWF imaging and DFE did not differ significantly. Conclusion: RRD extent on DFE and UWF images was consistent with intraoperative findings. UWF and DFE detection of peripheral retinal tears was similar, but 25% of retinal breaks were missed until intraoperative evaluation.

## 1. Introduction

The ophthalmoscope, developed by Hermann von Helmholtz in 1850, allows for visualization of the structures of the eye fundus and detailed studies of the vitreous, retina, and optic nerve head [1]. Subsequent advances in retinal imaging enabled photo-documentation for clinical purposes, research, telemedicine, and patient education during routine eye care, thereby improving both the understanding and management of vitreoretinal diseases [2].

For many years, most digital instruments for fundus photography captured images with only a 30 or 45° field of view. To obtain an image beyond the vascular arcades, photographs were obtained of the eyes during fixation at different points. For a wider vision of the retina, the photos were joined to create a composition or collage covering approximately 75° [3]. Although very useful, these images showed the retina just beyond the equator of the eye, providing no information about the retinal periphery.

Retinography of the anterior retina is difficult. Recent technologic advances, however, allow for imaging of the entire retina. Ultra-wide field (UWF) imaging is a rapidly evolving diagnostic modality that can image the peripheral retina to a greater extent than traditional fundus photography. The Optos 200Tx imaging system (Optos PLC, Dunfermline, Scotland, UK) is a confocal laser scanning ophthalmoscope with a parabolic mirror designed to obtain UWF images of the retina, up to 200° (internal angle; external angle of 120°). A single shot, acquired with or without mydriasis, provides a retinogram of more than 80% of the retinal surface [4,5]. Thus, simultaneous evaluation of the peripheral and central retina is possible without the need for significant eye steering. Visualization of the peripheral retina is essential for screening and diagnosing many vision-threatening diseases, as well as for monitoring treatment [6,7,8,9,10,11,12,13].

The gold standard for evaluating the peripheral retina is the dilated fundus examination (DFE) with scleral depression performed by a retina specialist. Documentation of retinal pathology, such as rhegmatogenous retinal detachment (RRD) or retinal tears, frequently relies on drawings. This method is time-consuming, requires an experienced retina specialist, active patient collaboration, and is uncomfortable for the patient. The purpose of this study was to compare the location and extent of RRD and retinal tears position evaluated by a non-mydriatic UWF fundus imaging, DFE, and intraoperative examination.

## 2. Materials and Methods

This was an observational retrospective chart review of patients operated on for primary RRD during 14 months in a vitreoretinal clinic at FISABIO Medical Ophthalmology Center (Valencia, Spain). The Research Ethics Committee (CEI of FISABIO) approved the study design and all methods adhered to the tenets of the Declaration of Helsinki.

Adult patients who presented with acute RRD during those dates and underwent surgical repair were reviewed. Only those who underwent UWF imaging were included. Participants without a detailed DFE (prior to surgery) or intraoperative examination by a retinal specialist, or with poor image quality were excluded from the study.

An Optos 200Tx UWF 200º retinal camera using green (532 nm) and red (633 nm) lasers was used to obtain fundus images in the primary position of gaze (PPG) by a single photographer without pupil dilation prior to surgery.

All patients underwent pre- and postoperative best-corrected visual acuity (BCVA) testing (decimal scale), slit lamp biomicroscopy, and DFE. All patients also underwent spectral-domain optical coherence tomography (SD-OCT) using the Spectralis HRA + OCT (Heidelberg Engineering Inc., Heidelberg, Germany).

A trained retina specialist masked to the clinical data independently analyzed the UWF fundus images for RRD quadrant location (nasal, temporal, superior, and inferior), macula on/off, and the presence and location of retinal breaks (anterior or posterior to equator). Then, a different retina specialist reviewed the medical records of patients looking for the same variables obtained during DFE and intraoperatively. Other data collected included demographics, presence of macular involvement based on SD-OCT, type of surgical repair, and the re-detachment rate. RRD surgeries were performed within 48 h after DFE and UWF imaging acquisition.

Comparison between UWF imaging, and clinical and intraoperative examination findings were made using the chi-square test. The association of UWF imaging with Lincoff’s rules for RRD was also analyzed. A p value of less than 0.05 was considered statistically significant.

## 3. Results

One hundred and twenty-three patients (123 eyes) with primary RRD were included in the final analysis. Their demographics and clinical features are summarized in Table 1. Briefly, mean age was 59.78 ± 14.91 years, 75.6% were male, and 57.4% were phakic at the time of surgery. The spherical equivalent was −2.00 ± 4.4 D (range: +5.25 D to −19.00 D).

Regarding the type of surgical repair, 18.7% underwent pars plana vitrectomy, 62.6% additionally underwent a scleral buckle, and 17.9% underwent an extrascleral procedure. The vitreous substitutions used for exchange were 54.6% sulfur hexafluoride (SF6) gas, 25.2% perfluoropropane (C3F8) gas, 17.6% air, and 2.5% silicon oil, depending on the RRD characteristics. The rate of re-detachment was 13.8%. The BCVA increased from 0.25 ± 0.3 decimal (20/80 Snellen) before surgery to 0.67 ± 0.3 decimal (20/30 Snellen) at 12 months (*p* = 0.009, paired t-test).

Table 2 shows the quadrant locations of the RRD based on UWF imaging, DFE, and intraoperative examination. Superior (Figure 1) and inferior (Figure 2) quadrants were more commonly affected, followed by temporal (Figure 3) and nasal quadrants. The description of the RRD did not differ among the three methods.

Overall, retinal breaks were observed in 96.7% of the patients based on intraoperative findings, compared with 73.2% as assessed by DFE (*p* < 0.001) and 65% as assessed by UWF fundus imaging (*p* < 0.001). No significant differences were detected between UWF imaging and DFE. Anterior retinal breaks were identified in 74% of the patients based on intraoperative examination, compared with 45.5% when assessed by DFE (*p* < 0.001) and 39% when assessed by UWF fundus imaging (*p* < 0.001). Assuming that intraoperative examination detected all the retinal breaks, UWF imaging in PPG missed 47.3% and DFE missed 38.5% of anterior retinal breaks. UWF imaging did not differ significantly from DFE. Posterior retinal breaks were detected in 28.5% of the patients during the intraoperative exam, 29.3% when assessed by DFE, and 30.9% when assessed by UWF fundus imaging. Detection of posterior retinal tears was equivalent among the three techniques with no significant difference between them. Retinal breaks observed in the intraoperative exam and not detected using UWF (Figure 4) imaging followed Lincoff rules in 70.5% of the cases.

Evaluation of the macula with OCT was not possible in 29.3% of the cases. Among those that were evaluated, 34.5% of the patients had macula-on retinal detachment, whereas 57.5% had macula-off retinal detachment, and 8% were not evaluable. Assessment of macular status by UWF imaging was possible in 121 patients (98.4%); 31.4% were estimated to have macula-on retinal detachment, 51.2% had macula-off retinal detachment, and 17.4% were not evaluable. No significant differences were detected between the two methods (macula-on, *p* = 0.33; macula-off, *p* = 0.19).

## 4. Discussion

RRD is a medical emergency. Proliferative vitreoretinopathy and blindness may result if left untreated or if surgery is delayed. Therefore, early diagnosis and treatment are essential for preserving visual function. DFE with scleral depression remains the gold standard for diagnosis and documentation of peripheral retinal lesions and RRD [14]. Many clinicians, however, do not regularly perform this clinical assessment. In addition, active patient collaboration for this procedure is critical and it is quite uncomfortable for the patient. Further, following the procedure, it is advisable to draw all the clinical findings, which is time-consuming, especially in the era of electronic medical records.

The results of this study confirmed that DFE and UWF imaging identify the extent of RRD consistently with intraoperative findings. UWF fundus imaging, however, has several advantages over other techniques: detailed 200º high-resolution images can be obtained in less than half a second and in a single shot, mydriasis is not required, and findings can be easily documented for clinical, teaching, or legal purposes. This makes it an ideal technique for areas where retina specialists are not readily available.

Kornberg et al. [15] compared UWF imaging and indirect ophthalmoscopy for detecting features of detachments in 34 patients. They found that UWF imaging improved detection of the extent of the RRD compared with DFE. By applying deep learning technology to images obtained from 407 patients, Ohsugi et al. [16] revealed that UWF fundus imaging has a sensitivity of 97.6% and a specificity of 96.5% for detecting RRD.

Early detection of anterior retinal breaks is crucial because missed retinal breaks are responsible for up to 64% of the cases of failed RRD surgery [17]. Kornberg et al. [15] also found that UWF imaging may be insufficient for detecting retinal breaks, especially in the far superior and inferior quadrants. Khandhadia et al. [18] reported a 33% sensitivity of UWF imaging for detecting tears and holes. The poor detection of retinal breaks is consistent with our results. We found that UWF fundus imaging in PPG allows for the detection of anterior retinal breaks in 52.7% of cases.

This work has several limitations. The retrospective nature of the study and perhaps a more complete RRD classification system could have been used. UWF imaging was performed in PPG. If eye steering was used to obtain the images, the percentage of retinal breaks detected would likely have been higher. Nevertheless, no significant differences were detected between DFE and UWF imaging. Regarding posterior retinal breaks, there were no differences among the three examinations. Mackenzie et al. [19] found that detecting lesions posterior to the equator using UWF imaging had a 74% sensitivity, while detecting lesions anterior to the equator had only 45% sensitivity.

Lincoff provided the essential guidelines, four rules, to find the primary break in RRD [20]. Localizing the break correctly in the detached retina is not an easy task to perform preoperatively. Retinal breaks not detected with UWF imaging followed Lincoff rules in 70.5% of the cases, which greatly facilitates the surgeon in the surgical planning.

Consistent with previous studies [21], our results showed more RRD in males and in right eyes. This small preponderance might suggest differences in ocular anatomy relating to gender and laterality.

The status of the macula is a significant factor for the final visual outcome in RRD and must be evaluated at the time of surgical repair. SD-OCT is an excellent tool for macular evaluation, but clear preoperative OCT measurements could not be obtained in up to 29% of the patients in the present study due to a bullous RRD or vitreous opacity. UWF imaging allows for simultaneous evaluation of the peripheral and central retina, and 98.4% of the patients were satisfactorily evaluated.

Another limitation of the study is that we only included patients with good-quality UWF images. Media opacity, pupil size, retinal pigment epithelium status, and centering of the eye, assessed clinically, are factors that may affect the image quality and lead to underestimating the location of RRD and number of retinal breaks. Moreover, according to our RRD protocol, all participants were instructed to keep a relative rest and to maintain the most appropriate head position to minimize the risk of increasing the area of the detached retina prior to surgery. Most of them were operated within 24 h after DFE and, exceptionally, within 48 h. Consequently, we assumed that RRD extent did not change between clinical examinations and intraoperative evaluation.

## 5. Conclusions

UWF imaging precisely documented the RRD extent, consistent with DFE and intraoperative findings. The detection of anterior retinal breaks in PPG, however, is suboptimal compared with intraoperative findings. The UWF technology is still expensive, but may have an important clinical role for diagnosis, surgical planning, and follow-up of RRD. Further studies are warranted to determine the utility of UWF imaging in the management of RRD and other retinal diseases.

## Figures and Tables

**Figure 1 brainsci-10-00521-f001:**
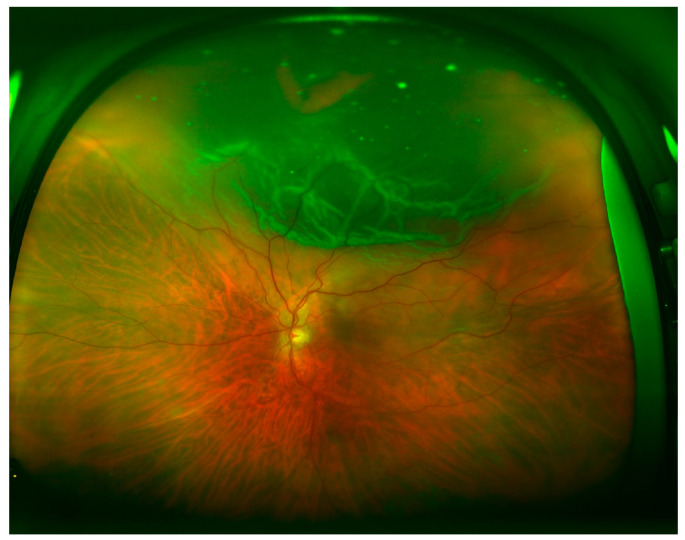
Superior rhegmatogenous retinal detachment (RRD), macula-off, and a horseshoe tear located at 12 o’clock.

**Figure 2 brainsci-10-00521-f002:**
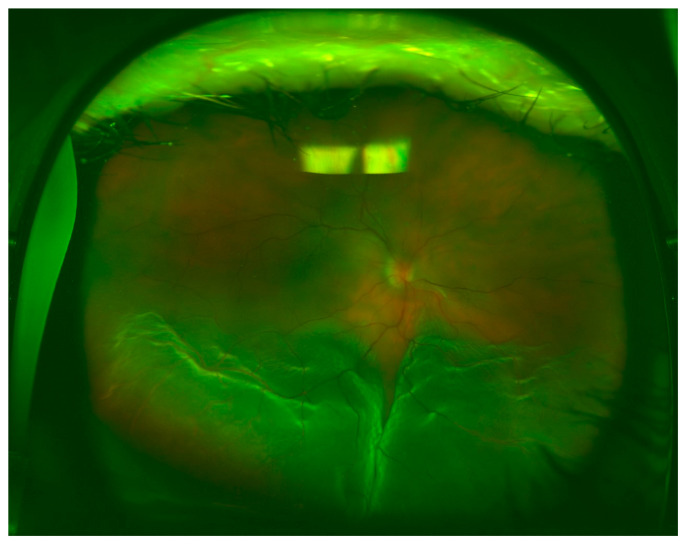
Inferior RRD, macula-off, and a round retinal hole located at 6 o’clock.

**Figure 3 brainsci-10-00521-f003:**
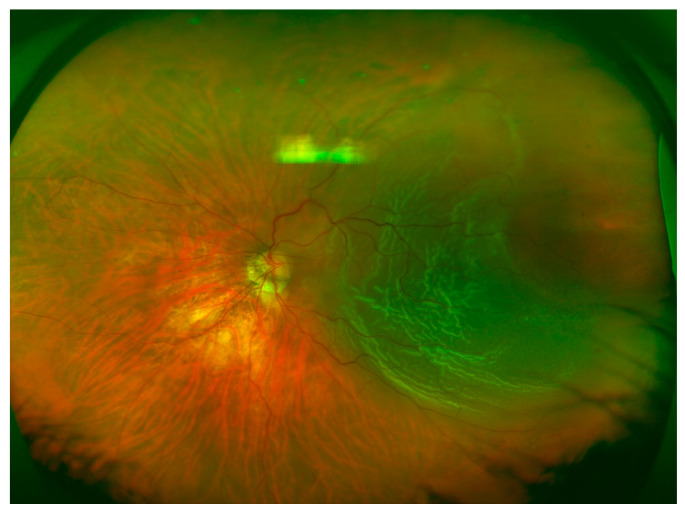
Temporal RRD, macula-off, and a retinal break located at 3 o’clock.

**Figure 4 brainsci-10-00521-f004:**
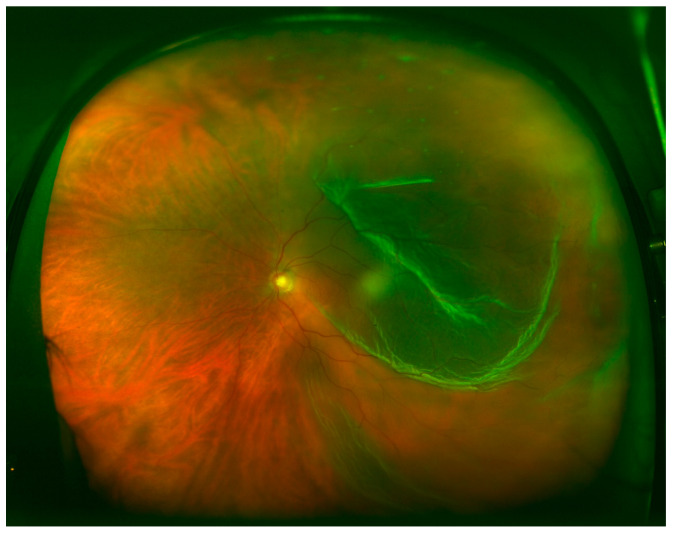
Temporal RRD, macula-off, and no retinal break observed.

**Table 1 brainsci-10-00521-t001:** Demographics and clinical characteristics of the sample.

	N or mean	% or ± SD
Age (years)	59.78	±14.91
Male	93	75.6%
Right eyes	62	57.9%
Phakic	70	57.4%
Refraction (D)	−2.00	±4.39
Preoperative decimal BCVA (Snellen)	0.25 (20/80)	±0.3
Fovea attached on SD-OCT	30	29.3%

N: number; %: percentage; D: diopters; BCVA: best corrected visual acuity; SD-OCT: spectral-domain optical coherence tomography.

**Table 2 brainsci-10-00521-t002:** Comparative assessment of location of RRD between UWF fundus imaging, DFE, and intraoperative evaluation.

	UWF Imaging	DFE	Intraoperative Evaluation
Superior	61 (49.6%)	57 (46.3%)	58 (47.2%)
Temporal	56 (45.5%)	51 (41.5%)	55 (44.7%)
Nasal	26 (21.1%)	24 (19.5%)	25 (20.3%)
Inferior	64 (52.0%)	53 (43.1%)	56 (45.5%)
Total	7 (5.7%)	11 (8.9%)	12 (9.8%)

UWF: ultra-wide field; DFE: dilated fundus exam.

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
