# Peer review of "Non-Mydriatic Ultra-Wide Field Imaging Versus Dilated Fundus Exam and Intraoperative Findings for Assessment of Rhegmatogenous Retinal Detachment"

_brainsci, 2020, doi:10.3390/brainsci10080521_

Round 1

Reviewer 1 Report

Interesting subject and highly relevant. Concept and design of the study are clear. The medico legal factors are also important. Some of the results could be made clearer to the reader. The assessment of RRD extent could be clarified. Did the patients classified as superior RRD also have inferior involvement as well? The results show that inferior RRD numbers were greater than superior which would be unusual in that pure inferior RDs (ie those with inferior SRF only and inferior pathology) are relatively uncommon. Perhaps a rigid RRD classification system could be used such as that used by BEAVRS database in the UK or other similar databases. The photographic examples given are good. I would be interested to see an example of an inferior RRD as well.  In the description of the demographics the authors state the number of males but the table gives the number and percentage of females. I think it would be best to stick to one or the other. This is a retrospective study which has its limitations. A prospective study in the future would be worthwhile.

Author Response

Thank you for your interesting comments and your valuable remarks. Please find our detailed responses below.

Interesting subject and highly relevant. Concept and design of the study are clear. The medico legal factors are also important. Some of the results could be made clearer to the reader. The assessment of RRD extent could be clarified. Did the patients classified as superior RRD also have inferior involvement as well?

We agree that we could have made a more detailed RRD classification and a comment has been added into the limitations of the study. Line 155-156: “This work has several limitations. The retrospective nature of the study and perhaps a more complete RRD classification system could have been used.”

Patients classified as superior RRD did not have any inferior involvement.

The results show that inferior RRD numbers were greater than superior which would be unusual in that pure inferior RDs (ie those with inferior SRF only and inferior pathology) are relatively uncommon.

We agree inferior RDs are relatively uncommon but our center provides tertiary eye-care services and perhaps that’s why we have more inferior RRD cases.

Perhaps a rigid RRD classification system could be used such as that used by BEAVRS database in the UK or other similar databases.

Thank you, we will consider it for future studies.

The photographic examples given are good. I would be interested to see an example of an inferior RRD as well. 

Thank you. An example of an inferior RRD (figure 2) has been added.

In the description of the demographics the authors state the number of males but the table gives the number and percentage of females. I think it would be best to stick to one or the other.

Thank you for your observation. The sentence has been corrected.

This is a retrospective study which has its limitations. A prospective study in the future would be worthwhile.

Thank you, we agree that a prospective and a more detailed RRD classification would be worthwhile. We have added this to the limitations of the study. Line 155-156: “This work has several limitations. The retrospective nature of the study and perhaps a more complete RRD classification system could have been used.”

Reviewer 2 Report

In this manuscript, the authors reported the comparison of non-mydriatic ultra-widefield imaging, dilated fundus exam, and intraoperative examination for the detection of rhegmatogenous retinal detachment. The authors showed that the location and extent of the rhegmatogenous retinal detachment and retinal tears position evaluated by these techniques were not significantly different. The utility of this technique needs to be further discussed. This reviewer has the following comments:   1. The authors introduced three different methods. However, the authors show only an ultra-widefield image. Is it possible to show the images of the dilated fundus exam and intraoperative examination? 2. Figures 1, 2, and 3 captions reported that a horseshoe tear, retinal break, and no retinal break. However, the authors did not outline the position of them. The reviewer suggests that the authors should show an outline of these images. It is better to combine these images into one figure. 3. Does gender affect the results? Is it different between the left and right eyes? 4. Line 178-182: font size is smaller than other paragraphs.

Author Response

First, thank you for your time and effort reviewing our manuscript.

In this manuscript, the authors reported the comparison of non-mydriatic ultra-widefield imaging, dilated fundus exam, and intraoperative examination for the detection of rhegmatogenous retinal detachment. The authors showed that the location and extent of the rhegmatogenous retinal detachment and retinal tears position evaluated by these techniques were not significantly different. The utility of this technique needs to be further discussed. This reviewer has the following comments:

The authors introduced three different methods. However, the authors show only an ultra-widefield image. Is it possible to show the images of the dilated fundus exam and intraoperative examination?

Thank you for your comment. It was a retrospective study and based on the patient's medical history, a detailed fundus examination and retinal drawing. Unfortunately, we do not have intraoperative or DFE images of the cases.

Figures 1, 2, and 3 captions reported that a horseshoe tear, retinal break, and no retinal break. However, the authors did not outline the position of them. The reviewer suggests that the authors should show an outline of these images. It is better to combine these images into one figure.

Thank you. The position of the tear is described in the legend of each figure. Reviewer #1 prefer to add one more figure. However, if you consider this change necessary, please let us know.

Does gender affect the results? Is it different between the left and right eyes?

Thank you for your question. Our study showed more RRD in males and in right eyes. These results are consistent with previous studies reporting a small preponderance of RRD in males and in right eyes. A brief sentence and a reference have been added to the discussion. Line 166-167: “Consistent with previous studies [20], our results showed more RRD in males and in right eyes. This small preponderance might suggest differences in ocular anatomy relating to sex and laterality.”

Line 178-182: font size is smaller than other paragraphs.

Thank you very much. The font size has been corrected.

Reviewer 3 Report

I think it's a paper written to perfection.

You described that the retinal breaks were confirmed “intraoperatively” by a retinal specialist. There may be a difference in the intraoperative detection rate of retinal tears between vitrectomy and scleral buckling. It would be good if the accuracy of the detection rate of retinal fissures by surgical technique was described.

Author Response

We appreciate your comments and suggestions.

I think it's a paper written to perfection. You described that the retinal breaks were confirmed “intraoperatively” by a retinal specialist. There may be a difference in the intraoperative detection rate of retinal tears between vitrectomy and scleral buckling. It would be good if the accuracy of the detection rate of retinal fissures by surgical technique was described.

Thank you very much for your comment. Due to the relatively small number of cases with extrascleral procedure alone we have not studied these differences. Nevertheless, if you feel that we have not adequately addressed this concern, we will be happy to perform this subanalysis with our number of cases.  We will certainly take into account this comment for a future study.